# Sex Differences in the Oxygenation of the Left and Right Prefrontal Cortex during Moderate-Intensity Exercise

**DOI:** 10.3390/ijerph18105212

**Published:** 2021-05-14

**Authors:** Yuta Inagaki, Reo Sato, Takashi Uchiyama, Sho Kojima, Shinichiro Morishita, Weixiang Qin, Atsuhiro Tsubaki

**Affiliations:** 1Department of Physical Therapy, Niigata University of Health and Welfare, Niigata 950-3198, Japan; hpa14046@nuhw.ac.jp (R.S.); hpa14012@nuhw.ac.jp (T.U.); hpa14034@nuhw.ac.jp (S.K.); morishita@nuhw.ac.jp (S.M.); tsubaki@nuhw.ac.jp (A.T.); 2Department of Rehabilitation, Kobe City Medical Center General Hospital, Kobe-city 650-0047, Japan; 3Institute for Human Movement and Medical Sciences, Niigata University of Health and Welfare, Niigata 950-3198, Japan; hpm17005@nuhw.ac.jp; 4Graduate School of Health and Welfare, Niigata University of Health and Welfare, Niigata 950-3198, Japan

**Keywords:** prefrontal cortex, oxygenated hemoglobin, deoxygenated hemoglobin, sex differences, aerobic exercise

## Abstract

Introduction: Differences in cognitive performance with exercise between men and women have previously been reported. In this study, we evaluated between-sex differences in oxygenation of the prefrontal cortex (PFC) with moderate-intensity aerobic exercise (AE), which could contribute to noted differences in cognitive function. Method: The subjects were ten men (age, 21.5 ± 0.5 years; height, 171.7 ± 4.8 cm; weight, 65.6 ± 5.6 kg) and ten women (age, 21.4 ± 0.5 years; height, 157.6 ± 4.9 cm; weight, 51.3 ± 6.5 kg). They completed our AE protocol, consisting of a 30-min leg-ergometer cycling at an intensity of 50% peak oxygen uptake, with an initial 4-min rest period for baseline measurement. Measures of the dynamics of cerebral oxygenation included: oxygenated hemoglobin (O2Hb) in the left and right PFC (LR-PFC) and deoxygenated hemoglobin (HHb). The 30-min exercise period was subdivided into six 5-min phases, with the average and peak values determined in each phase. Results: A significant interaction was found between LR-PFC HHb and sex (*p* < 0.001), with significantly higher values in men than in women in phases 3–6 (*p* < 0.05). Conclusion: We report a significant sex effect of HHb in the LR-PFC.

## 1. Introduction

Currently, medical practice provides appropriate treatment, rehabilitation, and exercise prescription to patients with a variety of diseases. One of the major interventions is aerobic exercise (AE). AE contributes to the maintenance and improvement of cardiopulmonary function, with ≥30-min of AE, performed at an intensity of 40–60% of peak oxygen uptake (peakVO_2_), being effective to improve physical activity levels [1]. It has been reported that regular AE can reduce body mass index (BMI) and lower resting heart rate among obese individuals [2,3]. In addition, AE is effective in reducing the prevalence of obesity [4]. Thus, AE is used for health promotion and anti-obesity treatment. Of note, moderate acute exercise improves not only physical function but also cognitive function through an increase in cerebral blood flow to the prefrontal cortex (PFC), which has been identified in both young and elderly individuals [5,6,7,8,9,10]. Based on current evidence, increased PFC oxygenation during AE has been shown to enhance physical and cognitive function. In addition, it has been clarified that the implementation of acute aerobic exercise for young men and women promotes induced neuroplasticity of the primary motor cortex and corticospinal pathway regardless of the difference in hormones between men and women [11]. Endocrine function associated with AE underpins improvements in adaptations to exercise and to improvement in cognitive and physical function, including optimal body composition [12,13]. However, unlike age, sex does not seem to be consistently taken into account for AE prescription. According to previous studies, testosterone is involved in muscle hypertrophy and improved performance with resistance exercise [14,15]. Increases in testosterone levels have also been reported after moderate-intensity AE [16,17,18]. In addition, testosterone has been reported to be involved in suppressing the onset of lifestyle-related diseases, such as diabetes [19], and low levels of testosterone are associated with more atherosclerosis, coronary artery disease, and cardiovascular events [20,21,22]. The effects of testosterone on increases in muscle strength have previously been reported, such as higher knee flexion and extension muscle strength, normalized to the proportion of thigh muscles, in men than in women [23]. However, interpretation of acute changes with AE are more difficult to interpret, due to significant changes in estrogen levels during the menstrual cycle, although some studies have reported increased estrogen levels with exercise [24,25]. Increases in estrogen levels have been associated to muscle protection during resistance exercise [26], as well as lower impedance to microcirculation in the brain and increases the blood flow in the internal carotid artery system [27]. Estrogen receptors are localized on vascular endothelial cells and vascular smooth muscle cells, causing a vasodilation response, an increase in cerebral blood flow in the cerebrum and cerebellum, and maintenance or improvement of cognitive function [28,29,30]. The improvements in cognitive function with AE are sex-specific, with a higher correct answer rate to visual tasks for men and to language tasks for women [31]. There may also be an age-specific effect, with a significant delay in blood flow in the region of the middle cerebral artery during medium-intensity AE having been reported in older women, compared to younger women [32]. A further study highlighted the sex-specific effect, regardless of age, with differences in cerebral perfusion pressure during exercise between younger men and women [33]. A previous study evaluating sex differences in the PFC demonstrated a significantly higher level of PFC oxygenation in men than in women during cognitive tasks [34]. However, other studies have reported that women were found to have greater cerebral oxygenation levels during cognitive tasks [35,36]. There are also reports that no significant differences were found between the sexes in this aspect [37]. This shows a discrepancy in current evidence regarding sex-based difference the oxygenation of the PFC during cognitive tasks. Despite knowledge of these sex differences for exercise prescription, the effects of AE on PFC oxygenation are inconsistent.

Near infrared spectroscopy (NIRS) provides the technology to measure the oxygen dynamics in the cerebral cortex during AE. NIRS measures the change in the concentration of oxygenated hemoglobin (O_2_Hb) and deoxygenated hemoglobin (HHb) caused by cerebral blood flow responses, coupled with nerve activity [38], with an increase in O_2_Hb reflecting an increase in the nerve activity in brain parenchyma [39]. NIRS measurements are robust, providing reliable measures even during unrestrained activities, such as running and leg-cycle ergometer pedaling [40,41]. As such, the purpose of our study was to investigate sex-specific differences in PFC oxygenation during moderate-intensity AE. In this study, we hypothesized that left and right PFC O_2_Hb showed a significantly greater increase with exercise in women than in men.

## 2. Materials and Methods

The relevant schematic representation of the study procedures and characteristics of our study population are shown in Figure 1 and Table 1. We enrolled 20 young adults, in their 20s (men, 21.5 ± 0.5 years; women, 21.4 ± 0.5 years), with no significant past medical history. The Edinburgh Handedness Inventory was used to determine dominance, with only right-handed individuals included in our study group. None of the participants enrolled had been habitually exercising over the previous 6 months. Participants were screened to exclude a history of respiratory, circulatory, and neurological diseases.

Our study was approved by the ethics committee of Niigata University of Health and Welfare (approval number: 17770-170106) and was conducted in accordance with the tenets of the Declaration of Helsinki.

A ramped cardiopulmonary exercise protocol was used to determine the peak VO_2_ for each participant, as previously described [42,43]. Briefly, the protocol consisted of a 4 min rest, followed by a 4 min warm-up, the ramped cardiopulmonary protocol, and a 2 min cool-down. For the ramped protocol, the workload was increased in workload increments workload of 20 W/min, using stationary bicycle (Aerobike 75XLIII; Konami, Tokyo, Japan). To measure the VO_2_, an exhaled gas analyzer was used (AE-310S; Minato Medical Science, Osaka, Japan). Exhaustion was defined as follows [44] (1): a plateau VO_2_; (2) a respiratory exchange ratio > 1.1; (3) HR values near the age-predicted maximal heart rate, calculated as 220 − (0.65 × age); and (4) a decrease in the cycling cadence to <50 rpm, despite strong verbal encouragement. The highest value VO_2_ obtained was considered as the peak VO_2_. Additionally, we determined anaerobic thresholds (ATs) using the V-slope method during the ramped protocol, as previously described [45].

The aerobic exercise protocol is shown in Figure 2. To measure PFC oxygenation in all participants, a leg-cycle ergometer was used for AE, at a constant load of 50% of the measured peak VO_2_ for each participant for a duration of 30 min, which was deemed to be a moderate-intensity level of AE. A schematic of the experimental protocol is provided in Figure 1. The session consisted of an initial 4-min resting period, with the subsequent AE period subdivided into six 5-min phases, during which participants were verbally encouraged to maintain a pedaling rate of 55–60 revolutions per min. To prevent movement artefacts on measurements, participants were instructed to not move their head or trunk and to not talk during measurements.

A brain oxygen monitor (OMM-3000/16, Shimadzu Corporation) was used for the measurement of the dynamics of cerebral oxygenation. The following variables were measured at baseline and during each of the six phases of the AE protocol: O_2_Hb of the left and right PFC; HHb of the left and right PFC; mean arterial pressure (MAP); and scalp blood flow (SBF). According to the international 10–20 system, the head holder was set along the Fz position, which was located at three-tenths of the distance from the glabella to the external occipital protuberance. Ten source fiber and ten detector fiber were applied in a 5 × 4 array of 5, and PFC oxygenation was measured using all 31 channels. The fiber spacing was set at 30 mm, with a sampling rate of 160 ms. Recording from the region of interest was captured as follows; channels 1, 4, 5, and 8 for the left PFC, and channels 3, 6, 7, and 10 for the right PFC (Figure 3). O_2_Hb and HHb data were averaged at each channel, and a 0.1-Hz low pass filter was used to decrease noise from the heartbeat [46,47]. As previously described [48,49], cerebral hemoglobin concentration was measured using three differential continuous waves (780 nm, 804 nm, and 830 nm), based on the Modified Beer–Lambert Law [50], with absorbance at the start of measurement was defined as the initial absorbance of each wavelength. As it was not possible to measure the differential path-lengths using the continuous-wave NIRS system, absorbance was assumed to be a constant, with changes in the hemoglobin signal denoted in arbitrary units (mM·cm) [51,52].

A previous study indicated that O_2_Hb is likely to be affected by the MAP and SBF [53]. Therefore, these latter two variables were also measured. MAP was measured using a Finometer (Finapres Medical Systems, Amsterdam, The Netherlands) attached to the third finger of the left hand, with SBF measured at the forehead using a laser blood flowmeter (Omegaflo FLO-C1; Omegawave Inc., Tokyo, Japan). MAP and SBF were measured at baseline and during each of the six phases of the AE protocol.

Each measure was calculated as the change in value during each of the six phases of the AE protocol, from baseline measured during the 4-min pre-exercise resting period. To calculate the mean and standard deviation (SD) values over each of the six phases of the AE protocol, each 5-min phase was subdivided into 1-min intervals. Variables were then compared between men and women using a two-way repeated-measures analysis of variance (ANOVA), with the Tukey–Kramer method used as a post-hoc test. The peak (highest) and time to peak value were then determined. Next, we performed a normality test and confirmed homoscedasticity. A student’s t-test was used to evaluate between-phase differences within each sex. All analyses were performed using Excel (Microsoft), with the add-in software Statcel 4 (OMS Publishing Inc., Saitama, Japan). A significance level of 0.05 was used for all statistical analyses.

## 3. Results

### 3.1. Changes in the Oxygenation Dynamics of the PFC during Exercise 

#### 3.1.1. Left and Right PFC Oxygen Dynamics

The results are shown in Table 2 and Table 3. The two-way repeated measures ANOVA for the O_2_Hb of the left PFC revealed a main effect of time (F_(6, 126)_ = 15.17, *p* < 0.001) and sex (F_(1, 126)_ = 28.52, *p* < 0.001), with no significant time by sex interaction (F_(6, 126)_ = 1.83, *p* = 0.097). Similarly for the right PFC, we identified a main effect of time (F_(6, 126)_ = 17.97, *p* < 0.001) and sex (F_(1, 126)_ = 25.05, *p* < 0.001), with no significant time by sex interaction (F_(6, 126)_ = 1.40, *p* = 0.21). For HHb, for the left PFC, we identified a main effect time (F_(6, 126)_ = 8.67, *p* < 0.001) and sex (F_(1, 126)_ = 37.80, *p* < 0.001), as well as a time by sex interaction (F_(6, 126)_ = 6.49, *p* < 0.001). The post-hoc analysis identified a significantly higher HHb value in phases 3–6 in men than in women (*p* < 0.05). In fact, for phase 4–6, a significant increase in HHb, compared to baseline, was only identified in men (*p* < 0.05). The findings were comparable for the right PFC, with a main effect of time (F_(6, 126)_ = 10.10, *p* < 0.001) and condition (F_(1, 126)_ = 28.29, *p* < 0.001) on HHb, as well as a time by condition interaction (F_(6, 126)_ = 3.48, *p* = 0.003). Again, on post-hoc analysis, HHb values were significantly greater in men than in women for phase 3–6 of the exercise protocol (*p* < 0.05). However, in the right PFC, while only men showed an increase in HHb level, compared to baseline, in phase 4–6 (*p* < 0.01), women also did show a significantly higher HHb level in phases 5–6, compared to baseline (*p* < 0.05).

#### 3.1.2. Changes in the MAP and SBF during Exercise

The results are shown in Table 4. MAP was influenced by a main effect of time (F_(6, 126)_ = 13.47, *p* < 0.001) but with no influence of sex (F_(1, 126)_ = 4.64, *p* = 0.033) or a time by sex interaction (F_(6, 126)_ = 19.00, *p* = 0.97). In contrast, SBF was influenced by a main effect of both time (F_(6, 126)_ = 19.00, *p* < 0.001) and sex (F_(1, 126)_ = 0.11, *p* < 0.74), without a time by sex interaction (F_(6, 126)_ = 0.03, *p* = 0.99).

### 3.2. Comparison of the Peak Time and Peak Value for Each Measure during Exercise 

The results are shown in Table 5. No sex differences were identified for peak time or peak values for any of the measured variables (paired *t*-test). Student’s *t*-test analysis revealed significantly higher peak values of O_2_Hb (*p* < 0.05) and HHb (*p* < 0.01) in men than in women, for both the left and right PFC. No significant between-sex differences in the peak value of MAP and SBF were identified.

## 4. Discussion

### 4.1. Change in Each Measure during Exercise, Compared to the Resting Mean

The purpose of this study was to investigate sex-specific effects on PFC oxygenation during moderate-intensity AE. We examined this issue by comparing the PFC oxygen dynamics during a leg-cycle ergometer protocol of moderate-intensity AE between men and women. In this study, no condition interaction was observed in O_2_Hb of PFC. HHb was significantly increased in phases 4–6 in men. It is hard to say that changes in HHb alone reflect the neural activity. Therefore, we believe that there was no sex difference in the activation of neural activity in PFC in this study. However, a significant increase in HHb was observed only in men, which may be due to the effect of sex hormones on the vascular system. Exercise has been reported to increase estrogen levels in women [24], with increased estrogen levels reducing the impedance of microcirculation in the brain and increasing the blood flow to the internal carotid artery system [25]. On the other hand, testosterone effects, which are more abundant in men, on the cerebrovascular system are unclear. Therefore, it is possible that sex hormones affect changes in HHb.

In this study, there was no significant change in PFC O_2_Hb between men and women, but HHb was significantly increased in men. We believe this is a novelty in this study, but further studies are needed to consider the effects of sex hormones. In the future, it will be necessary to clarify the cerebral oxygen dynamics during exercise in men and women in consideration of the measurement of sex hormone concentration. We note that we did not identify a main effect of sex on MAP and SBF, indicating that the blood pressure and sweating responses during exercise are similar between men and women and, thus, would not have contributed to measured differences in O_2_Hb. It is important to consider, however, a previous study which reported a lower level of sweating among women than men during high-intensity exercise [54]. This exercise intensity specific difference could be explained by the more efficient sweating capacity (defined as the effective sweating/whole body sweating amount × 100) in women than in men. An increase in exercise intensity raises the body temperature and, thus, promotes a sweating response [55], as well as an increase in blood flow through the external carotid artery system, both of which increase the sweating response [52]. Therefore, between-sex differences in sweating, and possibly in MAP and SBF, would be expressed at higher levels of AE than we used in our study.

### 4.2. Peak Time and Peak Value of Each Measure during Exercise

We did not identify significant between-sex differences in the peak time of each measure, indicative of a similar underlying pattern of control of measured variables in men and women. Of note was our finding that peak values of oxygenation in the left and right PFC occurred at about 20 min of exercise, indicating that about 20 min of moderate-intensity AE is necessary to derive benefits, regardless of sex. However, considering our findings of higher peak O_2_Hb and peak HHb values in men than in women, moderate-intensity AE may be a more effective stimulus in men than in women.

### 4.3. Limitations of This Study

This study demonstrated sex-related differences in the response of oxygen dynamics to the left and right PFC to moderate-intensity AE. However, we did not clarify the actual effects of sex differences in oxygenation on cognitive function and sex hormone during AE. Moreover, our study group consisted of healthy young adults. In the future, it will be necessary to investigate changes in PFC during AE in the elderly, taking sex hormone into consideration. Moreover, in this study, potentially confounding factors were not included in our analysis. In the future, it will be necessary to collect sufficient information on cerebral oxygen dynamics and sex-related characteristics, as well as considering confounding factors. A study investigating the correlation between changes in fMRI and NIRS during brain activity reported a strong correlation with O_2_Hb but no correlation with HHb [56]. Therefore, it is difficult to say that NIRS alone reflects neural activity in the state where only HHb shows changes as in the study. The use of functional magnetic resonance (MR) imaging and MR spectroscopy would be useful in this regard.

## 5. Conclusions

This study focused on sex differences in PFC oxygen dynamics during AE in healthy young adults. We identified significant main effects of sex on the HHb in the left and right PFC during exercise, with levels being significantly higher in men than in women. Considering the absence of a sex effect on O_2_Hb, MAP, and SBF, our findings are indicative of a higher oxygen consumption in the PFC in men than in women during moderate-intensity AE.

## Figures and Tables

**Figure 1 ijerph-18-05212-f001:**
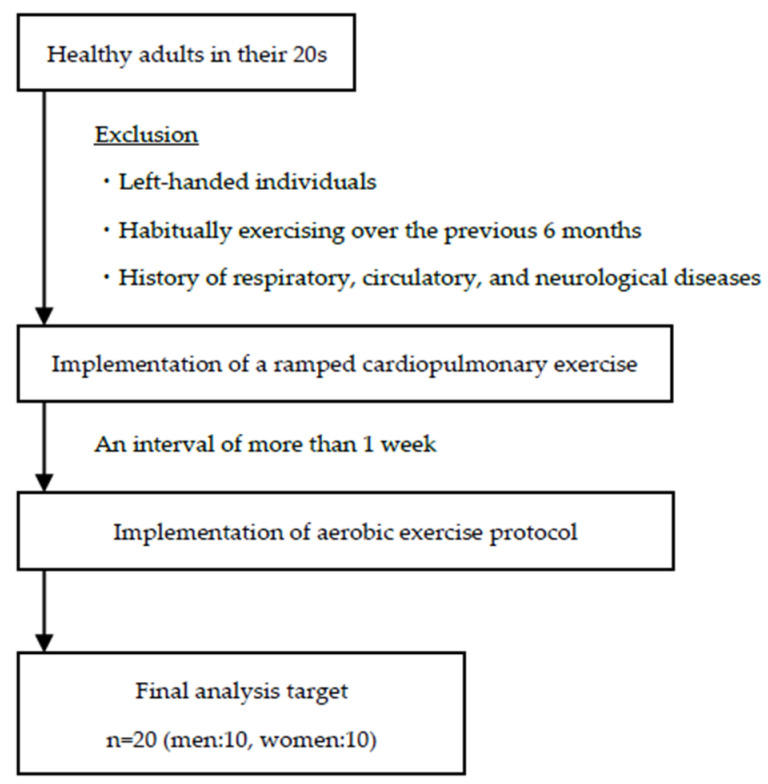
A schematic representation of the study procedures.

**Figure 2 ijerph-18-05212-f002:**
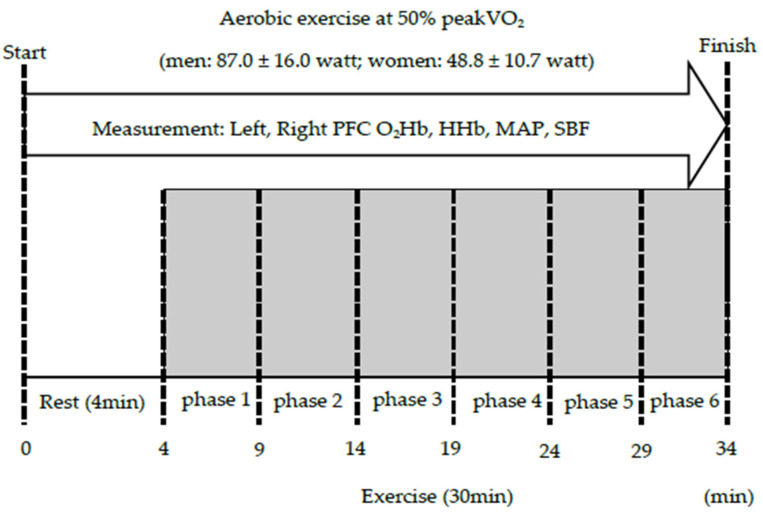
Experiment protocol. PFC: Prefrontal cortex, O_2_Hb: oxygenated hemoglobin, HHb: deoxygenated hemoglobin; MAP: Mean arterial pressure, SBF: Scalp blood flow.

**Figure 3 ijerph-18-05212-f003:**
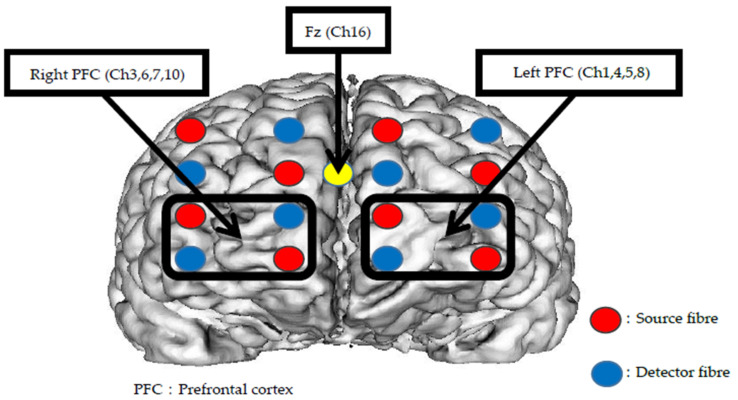
Measurement channels and fiber arrangement.

**Table 1 ijerph-18-05212-t001:** Participants’ characteristics.

	Men (*n* = 10)	Women (*n* = 10)	*p*-Value
Age	21.5 ± 0.5 (0.02)	21.4 ± 0.5 (0.02)	*p* = 0.67
Height (cm)	171.7 ± 4.8 (0.03)	157.6 ± 4.9 (0.03)	*p* < 0.001
Weight (kg)	65.6 ± 5.6 (0.09)	51.3 ± 6.5 (0.13)	*p* < 0.001
Peak VO_2_ (mL/min/kg)	1859.6 ± 244.5 (0.13)	1143.9 ± 351.0 (0.31)	*p* < 0.001
Peak load (watt)	189.7 ± 28.3 (0.15)	119.4 ± 19.0 (0.16)	*p* < 0.001
50% load (watt)	87.0 ± 16.0 (0.18)	48.8 ± 10.7 (0.22)	*p* < 0.001

Average ± standard error, ( ): CV range.

**Table 2 ijerph-18-05212-t002:** Changes in PFC O_2_Hb during exercise.

		Rest	Phase 1	Phase 2	Phase 3	Phase 4	Phase 5	Phase 6	Time	Between Sex	Interaction
Left PFC O_2_Hb (Mm·cm)	men	0.00 ± 0	0.00 ± 0.01	0.09 ± 0.02	0.13 ± 0.02	0.14 ± 0.02	0.14 ± 0.02	0.15 ± 0.02	*p* < 0.001	*p* < 0.001	*p* = 0.097
		(0)	(0)	(0.22)	(0.15)	(0.14)	(0.14)	(0.13)			
	women	0.00 ± 0	−0.01 ± 0.01	0.04 ± 0.01	0.06 ± 0.02	0.06 ± 0.03	0.07 ± 0.03	0.07 ± 0.03			
		(0)	(−1)	(0.25)	(0.33)	(0.5)	(0.43)	(0.43)			
Right PFC O_2_Hb (mM·cm)	men	0.00 ± 0	0.00 ± 0.01	0.07 ± 0.02	0.11 ± 0.02	0.11 ± 0.02	0.12 ± 0.02	0.12 ± 0.02	*p* < 0.001	*p* < 0.001	*p* = 0.21
		(0)	(0)	(0.29)	(0.18)	(0.18)	(0.17)	(0.17)			
	women	0.00 ± 0	−0.01 ± 0.01	0.04 ± 0.01	0.05 ± 0.01	0.06 ± 0.01	0.07 ± 0.01	0.06 ± 0.01			
		(0)	(−1)	(0.25)	(0.2)	(0.17)	(0.14)	(0.17)			

Average ± standard error, ( ): CV range; PFC: Prefrontal cortex, O_2_Hb: oxygenated hemoglobin

**Table 3 ijerph-18-05212-t003:** Changes in PFC *HHb* during exercise.

		Rest	Phase 1	Phase 2	Phase 3	Phase 4	Phase 5	Phase 6	Time	Between Sex	Interaction
Left PFC HHb (mM·cm)	men	0.00 ± 0	0.00 ± 0	0.01 ± 0.01	0.03 ± 0.01 †	0.05 ± 0.01 *,†	0.07 ± 0.01 **,††	0.08 ± 0.01 **,†	*p* < 0.001	*p* < 0.001	*p* < 0.001
		(0)	(0)	(1)	(0.33)	(0.2)	(0.14)	(0.13)			
	women	0.00 ± 0	0.01 ± 0	0.01 ± 0	0.00 ± 0	0.00 ± 0	0.01 ± 0.01	0.01 ± 0.01			
		(0)	(0)	(0)	(0)	(0)	(1)	(1)			
Right PFC HHb (mM·cm)	men	0.00 ± 0	0.01 ± 0	0.02 ± 0.01	0.04 ± 0.01	0.06 ± 0.01 **,†	0.07 ± 0.01 **,††	0.07 ± 0.02 **,††	*p* < 0.001	*p* < 0.001	*p* = 0.003
		(0)	(0)	(0.5)	(0.25)	(0.17)	(0.14)	(0.29)			
	women	0.00 ± 0	0.01 ± 0	0.01 ± 0.01	0.02 ± 0.01	0.02 ± 0	0.02 ± 0.00 *	0.03 ± 0.01 **			
		(0)	(0)	(1)	(0.5)	(0)	(0)	(0.33)			

Average ± standard error, ( ): CV range; vs rest *: *p* < 0.05, **: *p* < 0.01 vs women †: *p* < 0.05, ††: *p* < 0.01; PFC: Prefrontal cortex, HHb: deoxygenated hemoglobin.

**Table 4 ijerph-18-05212-t004:** Changes in MAP and SBF during exercise, from baseline.

		Rest	Phase 1	Phase 2	Phase 3	Phase 4	Phase 5	Phase 6	Time	Between Sex	Interaction
MAP (mmHg)	men	0.01 ± 0.02	25.05 ± 2.51	25.49 ± 3.88	18.73 ± 4.28	18.40 ± 4.22	17.13 ± 4.83	18.71 ± 3.53	*p* < 0.001	*p* = 0.033	*p* = 0.97
		(2.0)	(0.10)	(0.15)	(0.23)	(0.23)	(0.28)	(0.19)			
	women	−0.03 ± 0.06	21.68 ± 2.25	19.51 ± 1.64	14.65 ± 1.92	13.78 ± 2.07	14.38 ± 2.44	15.68 ± 2.84			
		(−2.0)	(0.10)	(0.08)	(0.13)	(0.15)	(0.17)	(0.18)			
SBF (a.u.)	men	0.01 ± 0	0.65 ± 0.39	3.74 ± 0.74	5.61 ± 1.02	6.46 ± 1.07	6.98 ± 1.17	6.99 ± 1.27	*p* < 0.001	*p* = 0.74	*p* = 0.99
		(0)	(0.6)	(0.20)	(0.18)	(0.17)	(0.17)	(0.18)			
	women	0.01 ± 0	0.08 ± 0.17	3.83 ± 1.07	5.32 ± 1.24	6.25 ± 1.28	7.01 ± 1.29	6.74 ± 1.13			
		(0)	(2.13)	(0.28)	(0.23)	(0.20)	(0.18)	(0.17)			

Average ± standard error, ( ): CV range; MAP: Mean arterial pressure, SBF: Scalp blood flow.

**Table 5 ijerph-18-05212-t005:** Peak time and peak value of each measure during exercise.

	**Peak Time (Men)**	**Peak Time (Women)**	***p*-Value**
Left PFC O_2_Hb	19.7 ± 2.5 (0.13)	17.8 ± 2.1 (0.12)	*p* = 0.56
Right PFC O_2_Hb	17.6 ± 2.0 (0.11)	20.3 ± 2.5 (0.12)	*p* = 0.40
Left PFC HHb	28.7 ± 0.5 (0.02)	22.8 ± 2.7 (0.12)	*p* = 0.084
Right PFC HHb	26.7 ± 1.1 (0.04)	24.3 ± 2.9 (0.12)	*p* = 0.45
MAP	6.5 ± 2.2 (0.34)	6.5 ± 2.4 (0.37)	*p* = 1
SBF	22.5 ± 1.6 (0.07)	22.1 ± 1.9 (0.09)	*p* = 0.88
	**Peak Value (Men)**	**Peak Value (Women)**	***p*-Value**
Left PFC O_2_Hb	0.17 ± 0.02 (0.12)	0.09 ± 0.02 (0.22)	*p* = 0.017
Right PFC O_2_Hb	0.15 ± 0.02 (0.13)	0.08 ± 0.01 (0.13)	*p* = 0.011
Left PFC HHb	0.08 ± 0.02 (0.25)	0.03 ± 0.01 (0.33)	*p* = 0.002
Right PFC HHb	0.09 ± 0.01 (0.11)	0.04 ± 0.01 (0.25)	*p* = 0.002
MAP	34.71 ± 3.52 (0.10)	29.98 ± 2.39 (0.08)	*p* = 0.27
SBF	8.14 ± 1.28 (0.16)	8.47 ± 1.42 (0.17)	*p* = 0.86

Average ± standard error, ( ): CV range; The unit of peak time is minutes; PFC: Prefrontal cortex, O_2_Hb: oxygenated hemoglobin, HHb: deoxygenated hemoglobin; MAP: Mean arterial pressure, SBF: Scalp blood flow.

## Data Availability

Not applicable.

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
