# Peer review of "Sex Differences in the Oxygenation of the Left and Right Prefrontal Cortex during Moderate-Intensity Exercise"

_ijerph, 2021, doi:10.3390/ijerph18105212_

Round 1

Reviewer 1 Report

few times as a reviewer I usually accept the document in its current form. However this is one of those occasions.

Methodologically it is perfectly described, and except for some grammatical and typographical errors, the document and its structure in all parts enjoy cohesion and structural coherence.

Although the findings have not been extremely significant, I think they show a very interesting line of research for the future.

My sincere congratulations to the authors

Author Response

Reviewer1

We have identified changes made to the text in response to comments and suggestions made in ‘red font’ text. We also provide a point-by-point response.

few times as a reviewer I usually accept the document in its current form. However this is one of those occasions.

Methodologically it is perfectly described, and except for some grammatical and typographical errors, the document and its structure in all parts enjoy cohesion and structural coherence.

Although the findings have not been extremely significant, I think they show a very interesting line of research for the future.

My sincere congratulations to the authors

[Authors’ reply]

 Thank you for your comment. We will continue to use this research as a basis for further studies.

Thank you for accepting the review.

Reviewer 2 Report

Table 4 was not found. Therefore, I can not review correctly. I recommend resubmission. 

Table 3 and 2 are reversed.

Figure 2 and 1 are reversed.

What is the mean of yellow color in "=" in page 5?

There was no discussion about significant sex difference in MAP (Table 3). How was resting MAP?

Author Response

Reviewer2

We have identified changes made to the text in response to comments and suggestions made in ‘red font’ text. We also provide a point-by-point response.

Table 4 was not found. Therefore, I can not review correctly. I recommend resubmission.

Table 3 and 2 are reversed.

Figure 2 and 1 are reversed.

What is the mean of yellow color in "=" in page 5?

[Authors’ reply]

Thank you for raising this issue. Regarding the above problem, it is possible that the position of the table or figure has changed or disappeared for some reason at the time of posting. Also, "=" has no particular meaning. We have corrected this in the manuscript.

There was no discussion about significant sex difference in MAP (Table 3). How was resting MAP?

[Authors’ reply]

 Thank you for your advice. In this study, we used a repeated measures two-way ANOVA. Since the MAP did not show a main effect on terms of interaction, we concluded that this study did not show any change due to sex differences, including the resting condition.

Reviewer 3 Report

  • The authors examined Sex Differences in the Oxygenation of the Left and Right Pre-frontal Cortex during Moderate-Intensity Exercise. This study is interesting, but there are some concerns.
  • There are many grammatical issues throughout the text that need to be fixed.

Abstract

  • The abstract needs to contain anthropometric characteristics of participants such as age, height, weight, etc.

Introduction

  • Lines 34-35: VO2 peak is not maximum oxygen uptake. Peak VO2 (VO2peak), directly reflective of VO2max, is the highest value of VO2 attained upon incremental or other high-intensity exercise tests designed to bring the subject to the limit of tolerance.
  • Lines 50-52: You need to include examples of more diseases in the following sentence:

“Furthermore, it has been reported that testosterone is involved in suppressing the onset of lifestyle-related diseases, such as diabetes [19].”

  • Lines 60-62: More references are needed to support the following sentence:

“Oestrogen receptors are localised on vascular endothelial cells and vascular smooth muscle cells, causing a vasodilation response, an increase in cerebral blood flow in the cerebrum and cerebellum, and maintenance or improvement of cognitive function [25].”

  • Lines 71-72: More references are needed to support the following sentence:

“However, in other studies, no significant sex differences were found [30]”

  • Lines 76-77: You need to provide a reference(s) to support the following sentence:

“Based on current evidence, increased PFC oxygenation during AE has been shown to facilitate the acquisition of physical and cognitive function.”

  • Lines 82-84: You need to provide a reference(s) to support the following sentence:

“NIRS measurements are robust, providing reliable measures even during unrestrained activities, such as running and leg-cycle ergometer pedalling.”

  • The study “Hypotheses” need to be included in the “Introduction”.

Methods

  • The inclusion and exclusion criteria need to be stated more clearly.
  • Day-to-day test reliability, CV range, and intraclass correlation coefficients for the assessments need to be included for ALL the assessments.
  • Suggestion: Add a schematic representation of the study procedures to the “Methods” section.

Results

  • If significant, all the p values in this section need to be mentioned in exact amounts, not just 0.05 and 0.01.

Discussion

  • The novelty in your study should be clearly mentioned in the “discussion”.

Author Response

Reviewer3

 We have identified changes made to the text in response to comments and suggestions made in ‘red font’ text. We also provide a point-by-point response. We also conducted proofreading by an English proofreading company.

Abstract

The abstract needs to contain anthropometric characteristics of participants such as age, height, weight, etc.

[Authors’ reply]

Thank you for your advice. The above information has been added to Lines 18-20.

Introduction

Lines 34-35: VO2 peak is not maximum oxygen uptake. Peak VO2 (VO2peak), directly reflective of VO2max, is the highest value of VO2 attained upon incremental or other high-intensity exercise tests designed to bring the subject to the limit of tolerance.

[Authors’ reply]

Thank you for your advice. I had made a mistake in the notation. The notation in Line 34 has been changed.

Lines 50-52: You need to include examples of more diseases in the following sentence:

“Furthermore, it has been reported that testosterone is involved in suppressing the onset of lifestyle-related diseases, such as diabetes [19].”

[Authors’ reply]

Thank you for your advice. We have added a report on the relationship between low level of testosterone and the cardiovascular system and added the relevant content at Lines 52-55.

[Reference]

[1] Hak, A., Witteman, J. Low levels of endogenous androgens increase the risk of atherosclerosis in elderly men: the Rotterdam study. J Clin Endocrinol Metab 2002, 87, 3632-3639.

[2] Rosano, G., Sheiban I. Low testosterone levels are associated with coronary artery disease in male patients with angina. Int J Impot Res 2007, 19, 176-182.

[3] Ohlsson, C., Barrett, E. High serum testosterone is associated with reduced risk of cardiovascular events in elderly men. The MrOS (Osteoporotic Fractures in Men) study in Sweden. J Am Coll Cardiol 2011, 58, 1674-1681.

Lines 60-62: More references are needed to support the following sentence:

“Oestrogen receptors are localised on vascular endothelial cells and vascular smooth muscle cells, causing a vasodilation response, an increase in cerebral blood flow in the cerebrum and cerebellum, and maintenance or improvement of cognitive function [25].”

[Authors’ reply]

Thank you for your advice. The following report has been added to the References.

[Reference]

[1] Robison, L., Gannon, O. Contributions of sex to cerebrovascular function and pathology. Brain Res 2019, 1710, 43-60.

[2] Murat, A., Arif, C. Effect of Aerodiol administration on cerebral blood flow volume in postmenopausal women. Maturitas 2005, 52, 127-133.

[3] Alison, B., Emily, B. Distinct cognitive effects of estrogen and progesterone in menopausal women. Psychoneuroendocrinology 2015, 59, 25-36.

Lines 71-72: More references are needed to support the following sentence:

“However, in other studies, no significant sex differences were found [30]”

[Authors’ reply]

Thank you for your advice. At present, we believe that there is no established difference in activation during cognitive tasks between sex. Before the revision, it was stated that men recognized the activation of PFC oxygenation during the cognitive task, but we came across a report stating that women recognized the activation of cerebral oxygenation. Therefore, the wording in the text has been changed and added to Lines 74-78.

[Reference]

[1] Ting, L., Qingming, L, Gender-specific hemodynamics in prefrontal cortex during a verbal working memory task by near-infrared spectroscopy. Behav Brain Res 2010, 209, 148-153.

[2] Jill, G., Matthew, J. Sex differences in prefrontal cortical brain activity during fMRI of auditory verbal working memory. Neuropsychology 2005, 19, 509-519.

Lines 76-77: You need to provide a reference(s) to support the following sentence:

“Based on current evidence, increased PFC oxygenation during AE has been shown to facilitate the acquisition of physical and cognitive function.”

[Authors’ reply]

Thank you for your advice. We thought that it was difficult to connect the context of the sentence you pointed out. Therefore, considering the context, we changed a similar sentence in Lines 41-43.

Lines 82-84: You need to provide a reference(s) to support the following sentence:

“NIRS measurements are robust, providing reliable measures even during unrestrained activities, such as running and leg-cycle ergometer pedalling.”

[Authors’ reply]

Thank you for your advice. The following related papers have been cited and added to the References.

[Reference]

[1] Zhiguang, J., Tian, F. Influence of acute combined physical and cognitive exercise on cognitive function: an NIRS study. Peer J 2019, 7, e7418.

[2] Tsubaki, A., Takai, H. Changes in Cortical Oxyhaemoglobin Signal During Low-Intensity Cycle Ergometer Activity: A Near-Infrared Spectroscopy Study. Adv Exp Med Biol 2016, 876, 79-85.

The study “Hypotheses” need to be included in the “Introduction”.

[Authors’ reply]

Thank you for your advice. In this study, referring to previous studies, we thought that the cerebrovascular reaction during exercise would work more significantly in women than in men, and it would be easier to recognize the activation of oxygenation. Therefore, we hypothesized that a significant increase in oxygenated hemoglobin might be observed in women. We have added the content about the hypothesis in Lines 88-89.

Methods

The inclusion and exclusion criteria need to be stated more clearly.

[Authors’ reply]

Thank you for your advice. Regarding the exclusion criteria, it is stipulated that those who have continuous exercise habits and those who have respiratory, circulatory, or neurological disorders are excluded from the study subjects within 6 months. In this study, we wanted to clarify the relationship between cerebral oxygenation and sex, so we limited the collection of information. Therefore, we have created a research design so as not to set many exclusion criteria.

Day-to-day test reliability, CV range, and intraclass correlation coefficients for the assessments need to be included for ALL the assessments.

[Authors’ reply]

Thank you for your advice. We have added the CV range in Tables 1-4. The reproducibility of intraclass correlation coefficients has been shown from past reports. However, we have not confirmed it because we thought that it was different from the original purpose of this research. Below are two reports on reproducibility.

[Reference]

[1] van, de, Ven, M, J., Colier, W, N. Can cerebral blood volume be measured reproducibly with an improved near infrared spectroscopy system? J Cereb Blood Flow Metab 2001, 21, 110-113.

[2] Firbank, M., Elwell, C, E. Experimental and theoretical comparison of NIR spectroscopy measurements of cerebral hemoglobin changes. J Appl Physiol 1998, 85, 1915-1921.

Suggestion: Add a schematic representation of the study procedures to the “Methods” section.

[Authors’ reply]

Thank you for your advice. A schematic representation has been added as Figure 1.

Results

If significant, all the p values in this section need to be mentioned in exact amounts, not just 0.05 and 0.01.

[Authors’ reply]

Thank you for your advice. Considering significant figures, the specific P-values in each table are listed. For those with a P-value below 0.001, the description has been changed to p <0.001. In addition, the notation in Line 163-182 has been changed.

Discussion

The novelty in your study should be clearly mentioned in the “discussion”.

[Authors’ reply]

Thank you for your advice. We have added a sentence about novelty on Lines 207-209 for consideration.